# Prevalent genetic alterations in pediatric thyroid carcinoma: Insights from an Argentinean study

Sandra Lorena Colli[1☯], Marisa Esther Boycho[2☯], Patricia Papendieck[3],
Franco Mauricio Mangone[2], Ana Chiesa[3], Mercedes García Lombardi[4], Martín Medín[1],
Elena Noemí De Matteo[1], Mario Alejandro Lorenzetti[2‡*], María Victoria Preciado[2‡*]

1 División Patología, Hospital de Niños "Dr. Ricardo Gutiérrez", Buenos Aires, Argentina, 2 Laboratorio de Biología Molecular, División Patología, Instituto Multidisciplinario de Investigaciones en Patologías Pediátricas (IMIPP), CONICET-GCBA, Hospital de Niños "Dr. Ricardo Gutiérrez", Buenos Aires, Argentina, 3 División de Endocrinología, Centro de Investigaciones Endocrinológicas "Dr. César Bergadá" (CEDIE), CONICET-FEI-GCBA, Hospital de Niños "Dr. Ricardo Gutiérrez", Buenos Aires, Argentina, 4 Unidad de Oncología, Hospital de Niños "Dr. Ricardo Gutiérrez", Buenos Aires, Argentina

☯ These authors contributed equally to this work.
‡ MAL and MVP also contributed equally to this work.
* mvpreciado67@gmail.com (MVP); lorenzetticonicet@gmail.com (MAL)

## Abstract

Thyroid cancer is the primary endocrine malignancy, exhibiting distinct genomic drivers. The frequency of genetic alterations varies between adult and pediatric groups and across geographic regions and ethnicities. Molecular markers may serve as prognostic tools and/or specific treatment-selection tools in papillary thyroid carcinoma (PTC) in children. We sought to characterize molecular alterations in pediatric PTC from Argentina and test a future laboratory algorithm for molecular diagnosis and stratification. Immunohistochemistry, fluorescence in situ hybridization, and Sanger sequencing were performed on 57 pediatric PTC samples. This study assesses multiple genetic alterations, including fusions in RET, ALK, MET, BRAF, and NTRK genes, as well as the BRAF V600E single nucleotide variant. Fusions in known oncogenes were observed in 29.8% of cases (6 in RET, 5 in ALK, 4 in NTRK3, 1 in BRAF, and 1 in MET). The BRAF V600E SNV was detected in 12.3% of cases. Larger tumor size and higher initial risk were associated with genetic alterations ($P = 0.027$ and $P = 0.036$, respectively). Designing a laboratory algorithm following an increasing order of complexity provided a reliable molecular testing platform that reduces the requirement for NGS screening. These results also broaden the data on PTC alterations in children from Argentina.

## Introduction

Thyroid cancer (TC), the most common endocrine malignancy, shows increasing incidence across all age groups over the past decade [1]; however, its incidence in the pediatric population is still lower than in adults [2,3]. Within the pediatric

**Data availability statement:** The datasets generated and/or analyzed during the current study are available in Repositorio Institucional CONICET Digital following the URL: http://hdl.handle.net/11336/254857.

**Funding:** This study was supported by a grant from the National Institute of Cancer (INC) of the National Ministry of Health (Res 83/2020), grants from the National Agency for Research Promotion, Technological Development and Innovation (Agencia I+D+I) of the National Ministry of Science, Technology and Innovation (PICT 2020 No. 854) and (PICT 2021 No. 121) and a grant from Fundación Hector A. Barceló, year 2021-2022. There was no additional external funding received for this study. E.N.DM, A.C., M.A.L, and M.V.P are members of the CONICET Research Career Program. M.E.B was supported by a PhD fellowship from Agencia I+D+I.

**Competing interests:** The authors have declared that no competing interests exist.

group, the incidence of TC increases with age; in fact in most countries the age-standardized incidence rates (ASR) for TC, as published by the International Agency for Research on Cancer (IARC), increase around 10-fold between children (0–14 years) and adolescents (15–19 years) (S1 Table) [3,4]. Indeed, in Argentina the global ASR for pediatric cases (0–19 years) is 0.43 per 100000, but remarkably, the fold change between the ASR in children and adolescents increases by 34-fold (0.05 vs 1.7) [3]. The ASR value for children under 14 years of age reported by IARC, is consistent with the 0.04 per 100000 ASR reported in the *Registro Oncopediátrico Hospitalario Argentino* (ROHA) between 2000 and 2009 [5]. Moreover, IARC's ASR values denote that TC is the most prevalent cancer in the female adolescents group in America and Europe [3,4], a fact that highlights the importance of developing accurate protocols for the evaluation and management of pediatric cases of thyroid cancers.

In this regard, treatment guidelines tend towards a more personalized approach of pediatric patient with TC, considering not only the initial mortality risk, as assessed with the histological classification following the guide by the College of American Pathologists (CAP) and pTNM pathologic stage classification by the American Joint Committee on Cancer (AJCC) [6], but also predicting the initial risk assessment of recurrence as proposed by the American Thyroid Association (ATA) [7]. These guidelines classify patients into three initial risk categories from lower to higher risk, enabling to adjust for vigilance frequency or additional treatment. Additionally, a dynamic risk stratification [8] assesses initial treatment response by combining clinical and laboratory data produced after initial treatment and during follow-up, which allows for risk re-categorization after each medical evaluation and enables a more accurate management and follow-up of the patient, implying less frequent interventions in low risk cases [9].

A further step towards tailored medicine in TC in recent years involves the assessment of molecular markers, either chromosomal aberrations or single nucleotide variants (SNVs) that may assist as diagnostic, prognostic and/or serve as specific treatment-selection tools. Up to date, these molecular markers, some of them considered as TC drivers, include somatic SNVs in *BRAF* and *RAS* genes and gene fusions involving tyrosine kinase receptor genes *NTRK1*, *NTRK2*, *NTRK3*, *RET*, *BRAF*, *ALK* and *MET*. Regarding differentiated thyroid carcinomas (DTC) of the papillary thyroid cancer (PTC) histologic type, which accounts for~90% of pediatric TC, molecular markers are similar in children and adults. However, these vary with age; while SNVs are more common in adulthood (~70%) than in childhood (~30%), gene fusions are more prevalent in children (~40%) than in adults (~10%) [2,10–13] (S2 Table). Gene fusions primarily comprising *RET*, *NTRK1* or *NTRK3* and more rarely *ALK* or *MET* genes are among the most commonly described as tumor drivers in pediatric PTC [2,14]. More conflicting data exist regarding *BRAF* gene fusions, where some authors demonstrated higher prevalence of *AGK-BRAF*, especially in children < 10 years of age, but a lower incidence in children > 10 years of age ([15] and references herein). In point of fact, *BRAF*-driven thyroid carcinogenesis appears to vary with age and ethnicity; while *RET*, *NTRK* and *BRAF* fusions are primarily

described in Caucasian children, the prevalence of *BRAF* fusions decreases with age and *BRAF* SNVs, namely BRAF V600E, becomes more prevalent in adolescents and adult Hispanic populations [2,14,15].

Despite the growing body of evidence regarding the genomic profile of pediatric thyroid cancer, the applicability of these molecular markers is currently under research and validation. As a matter of fact, most of the current data come from European and North American Countries, but with more limited data from Asia [13,16] and even less from South America [10]. In this context, the aim of this study was to characterize molecular alterations in a series of pediatric cases of PTC from a reference pediatric hospital in Argentina and assess their frequency and association with clinical data at diagnosis and with comorbidities as well as their usefulness as prognostic or treatment decision tools.

## Materials and methods

### Ethics statement

Ricardo Gutiérrez Children Hospital's ethics committee, *Comité de Ética en Investigación* (CEI), has reviewed and approved this study (Internal CEI registry number N˚ 20.59, Institutional registry code 3120), which is in accordance with the human experimentation guidelines of our institution and with the Helsinki Declaration of 1975, as revised in 1983. A written informed consent was obtained from patients and patients guardians. Data were last accessed for research purposes in February 1st 2024; and in accordance with the Argentinean *Habeas Data* Law No. 25.326, authors had no access to information that could identify individual participants during or after data collection.

### Patients and samples

Pediatric patients, under 19 years of age, with thyroid tumors who were diagnosed, underwent surgery, received treatment and follow-up at Ricardo Gutiérrez Children Hospital between 2018 and 2022 were enrolled. Data were last accessed for research purposes in February 1st 2024. All patients underwent complete thyroidectomy and central compartment clearance. Radioactive iodine therapy was indicated as recommended in ATA guidelines [7]. A first intrasurgical frozen tissue section was examined for diagnostic purposes; subsequently, the full thyroidectomy piece was formalin-fixed and paraffin embedded (FFPE). Those participants with insufficient biopsy material were excluded from this study. Histological sections were blindly evaluated by two independent pathologists (SLC and MM). After routine macro and micro histological analysis, cases were classified and staged according to the guidelines introduced by the College of American Pathologists [17]. Postoperative staging according to ATA guidelines was performed between weeks 3 and 12 following surgery, allowing for initial stratification of patients in low, intermediate and high risk groups. Intermediate and high risk group patients underwent 131I therapy. After that, levothyroxine suppression therapy was initiated and follow up was indicated according to ATA guidelines [7]. Response to initial treatment was assessed at 12 months after surgery according to the ATA dynamic risk stratification system for adults [8].

### Immunohistochemical analysis

Four micrometer thick FFPE tissue slides were stained in an automated BenchMark XT instrument with antibodies anti-pan-TRK (Clone EPR17341) and anti-BRAF V600E (Clone VE1) following manufacturer's instructions for antigen retrieval, antibody incubation and detection with OptiView DAB IHC v5 reagents kit; all reagents and equipment from Ventana (Ventana-Roche, Rotkreuz, Switzerland). Each slide contained a previously characterized positive tissue section for each assayed antibody as a positive control. Positive staining with pan-TRK antibody in ≥ 1% of tumor cells, regardless of the intensity and staining pattern, was interpreted as a genetic alteration in *NTRK1*, *NTRK2* or *NTRK3* as described in [18].

### Fluorescent in situ Hybridization (FISH)

Four micrometer thick FFPE tissue slides were assayed for abnormal gene translocation by FISH in all cases. Large genetic alterations in *MET, BRAF*, *ALK*, *RET*, *ETV6*, *NTRK1*, *NTRK2* and *NTRK3* genes were assessed with specific



break apart probes (Lexel, Buenos Aires Argentina and CytoCell, Cambridge, UK) (S3 Table). In particular for *NTRK* genes, a sequential algorithm was applied; samples were first tested for *NTRK3* and when negative, tested for *NTRK1* and then *NTRK2*.

Briefly, deparaffinization was performed with xylene for 15 minutes, twice. Antigen retrieval was performed with sodium citrate buffer pH 6 for 35 minutes at 85°C. Slides were rinsed once with phosphate saline buffer (PBS) 1x for 5 minutes and tissue sections were digested with porcine stomach mucosa pepsin (Sigma, St. Louis, USA) solution (0.1gr/ml) for 17 minutes at 37°C, and rinsed twice in PBS. Tissue sections were dehydrated with a 70% – 100% ethanol series. Probes (10 µl) were then applied and the sections were covered with a coverslip and edges were sealed with rubber cement. Tissues were denatured at 75°C for 5 minutes, followed by incubation at 37°C overnight in a ThermoBrite incubator (Abbott Molecular, Des Plaines, USA). Slides were then washed with 2x citrate saline solution (CSS)/0.3% NP-40 at 72°C for 1–2 minutes, followed by a second rinse with 2x CSS at room temperature for 2 minutes. Nuclei were counterstained with 4',6-diamidino-2-phenylindole (DAPI), and fluorescence was visualized in an AxioScopeA1 epi-fluorescence microscope (Carl-Zeiss, Oberkochen, Germany) with dual color pass filter set 25 for FITC/rhodamine (Carl-Zeiss, Oberkochen, Germany). A minimum of 100 tumor nuclei were evaluated and specimens were considered positive for chromosome rearrangements when green and red signals were visible and separated by more than 1 signal diameter in at least one allele, in no less than 5% of tumor cells [19]. All probes were internally validated using two tissue slides which were known to contain the specific abnormality. Each FISH assay included two additional tissue slides; one with a previously characterized positive tissue section and one with a negative case to serve as positive and negative controls, respectively. Additionally, lack of positive signals in non-tumor marginal tissue was used as an internal negative control for each studied biopsy. Images were acquired with an Axiocam 503 color dedicated camera (Carl-Zeiss, Oberkochen, Germany).

## BRAF V600E molecular analysis

Material from FFPE blocks was amplified by PCR and directly sequenced by Sanger method to detect this relevant single nucleotide variant (SNV). DNA purification, PCR amplification for exon 15 of *BRAF* gene and sequencing reactions were performed as described in [20].

## Next Generation Sequencing (NGS)

Total DNA was purified from up to 3 20 µm thick FFPE tissue sections with High Pure PCR Template Preparation Kit (Roche, Rotkreuz, Switzerland) according to the manufacturer's instructions. In an independent reaction, total RNA was purified from other 3 FFPE tissue sections with RecoverAll Total Nucleic Acid Isolation Kit (Invitrogen, Waltham, USA) according to the manufacturer's instructions. Purified DNA and RNA were quantified in a NanoDrop One instrument (Thermo Fisher Scientific, Waltham, USA) and the relation between absorbance at 260 and 280 nm was assessed; samples with a 260/280 relation above 1.8 were considered adequate for library preparation. NGS sequencing was performed as a third party service at Argenetics Laboratories with the Oncomine Focus Assay panel, which allows concurrent analysis of DNA and RNA to simultaneously detect SNVs, insertions and deletions, copy number variations, and gene fusions, in a single workflow. Libraries were sequenced in an Ion GeneStudio S5 Plus sequencer and analyzed with the Ion Torrent Oncomine reporter platform (Thermo Fisher Scientific, Waltham, USA).

## Statistical analysis

Chi-squared test ($\chi$2-test) or Fisher's Exact test were used to examine the association between categorical variables and the Student's t-test was used to compare continuous variables which follow a Gaussian distribution. All tests were performed with GraphPad Prism v5.01 (GraphPad Software, La Jolla, USA). Benjamini-Hochberg false discovery rate (FDR) test for multiple comparison corrections was applied to statistically significant biologically related comparisons. No

 

correction was applied to other comparisons, as each addressed independent clinical questions. *P*-values < 0.05 were considered significant. Given the sample size constraints, the achieved statistical power was 70%. Multivariate logistic regression analysis was performed to account for potential confounding variables between the occurrence of genetic alterations and clinical variables such as, tumor size, initial ATA risk, histological variant, sex, and age at diagnosis. *P*-values were calculated using the Wald test and odds ratios (OR) were derived to quantify effect sizes. Calculations were run with statsmodels 0.14.4 library in python v3.10.

## Results

### Patients

Given the scarcity of knowledge in the South American population of children with PTC regarding the frequency of genetic alterations and how these impact on risk stratification, we sought to shed some light over this still-in-validation issue. In this work we report the incidence of PTC between the years 2018 and 2022, in a single reference pediatric hospital from Argentina, with a special consideration on the characterization of molecular alterations and their association with disease severity at diagnosis and dynamic risk stratification during follow-up. In this work, we included 58 pediatric patients with a median age of 13 years (range 7–18 years) and 79% females. The present series included 1 poorly differentiated and 57 PTC, where the classic variant was the predominant one in 25/57 cases (43.6%). Clinical and pathological features are described in Table 1. Additionally, 24/57 (42.1%) patients had thyroid autoimmune disease, while 3 patients had concomitant Graves disease and 2 other patients had Ataxia-telangiectasia. On the other hand, 5 cases had other neoplastic conditions. One patient had a previous acute lymphoid leukemia (ALL) without radiation treatment, 2 patients underwent radiation treatment prior to PTC diagnosis, one for medulloblastoma, while the other for suprarenal neuroblastoma. The other 2 remaining cases developed a synchronic neoplasia (melanoma and solid pseudopapillary pancreatic neoplasm).

As shown in Table 2, when only considering PTC, 30 were stratified as high risk, 20 as intermediate risk and 7 as low risk tumors according to initial ATA risk stratification system and no association was found between initial risk and histological variant. For those cases with at least one year of follow-up (53/57) the dynamic risk was assessed. Overall, after 1 year of follow-up, 30/53 (56.65%) cases achieved an excellent response (ER), 5/53 (11.3%) an indeterminate response (IR), 4/53 (7.6%) a biochemical incomplete response (BIR) and 13/53 (24.5%) a structural incomplete response (SIR). Additionally, the poorly differentiated carcinoma presented with high initial risk and achieved ER after a year of follow-up.

### Gene fusions and BRAF V600E variant

For the purpose of this study, and to test a future molecular platform for thyroid cancer diagnosis and risk stratification, all cases were assessed for all mentioned genetic alterations by multiple techniques. At first all cases were tested with pan-TRK by IHC and positive pan-TRK cases were tested by FISH with break-apart probes for *NTRK3*, *NTRK1* and *NTRK2*, in the specified sequential order given their fusion frequencies as a complementary method. Independent FISH assessment for *ETV6* rearrangements was performed when *NTRK3* rearrangements were found. In parallel, all cases were tested for BRAF V600E mutation by IHC as a rapid method, and by specific PCR amplification followed by Sanger sequencing. Additionally, all cases were further assessed for large rearrangements in *ALK*, *RET*, *MET* and *BRAF* genes by FISH independently of previous results. Cases still posing a diagnosis challenge were further analyzed by NGS sequencing.

Regarding large gene fusions, 17/57 (29.8%) cases of PTC harbored gene fusions in known oncogenes by FISH: 6 *RET,* 5 in *ALK*, 4 in *NTRK3*, 1 *BRAF* and 1 *MET* (Fig 1A and 1B). Concerning BRAF V600E expression, it was detected in 6/57 cases by IHC (Fig 1C) and, in parallel, *BRAF* c.1799A>T (p.V600E) point mutation was detected in 7/57 (12.3%) cases by Sanger sequencing (Fig 1D). Presumably, the lack of V600E IHC detection in one case could have been due to excessive FFPE tissue block weathering leaving scarce marginal tumoral tissue. All V600E cases were of the classic histologic variant and 5 were assessed as high initial risk and the remaining 2 cases as intermediate initial risk. All these alterations were mutually exclusive.



**Table 1. Clinical and pathological features of a pediatric cohort with papillary thyroid carcinoma (PTC).**

| Clinical Characteristics | All patients (n = 57) | Patients with no genetic alterations (n = 33) | Patients with genetic alterations (gene fusions + BRAF V600E) (n = 24) | P value [b] |
|---|---|---|---|---|
| **Histological variant** | | | | |
| Classic | 25 | 13 | 12 | 0.160 [c] |
| Follicular | 14 | 7 | 7 | |
| Diffuse Sclerosing | 6 | 4 | 2 | |
| Solid/Trabecular | 1 | 1 | 0 | |
| Warthin-like | 1 | 1 | 0 | |
| Oncocytic | 1 | 0 | 1 | |
| Microcarcinoma | 9 | 7 | 2 | |
| **Clinicopathological data** | | | | |
| Mean age ± SD (years) | 12.78 ± 2.68 | 12.61 ± 2.52 | 13.03 ± 2.94 | 0.565 |
| Sex (Female:Male) | 46:11 | 28:5 | 18:6 | 0.499 |
| Thyroid autoimmune disease | 24 | 16 | 8 | 0.420 |
| Tumor size (pTNM T1 + T2) | 37 | 23 | 14 | 0.027 [d] |
| Tumor size (pTNM T3 + T4) | 20 | 10 | 10 | |
| Tumor multifocality | 30 | 16 | 14 | 0.592 |
| Lymph vascular invasion | 35 | 20 | 15 | 1.000 |
| Lymph node metastasis | 39 | 20 | 19 | 0.161 |
| Distant metastasis | 9 | 4 | 5 | 0.471 |
| **Initial risk assessment** | | | | |
| Low | 7 | 7 | 0 | 0.036 [d] |
| Intermediate + High | 50 | 26 | 24 | |
| **Dynamic risk assessment [a]** | | | | |
| Excellent response (ER) | 30 | 18 | 12 | 0.589 |
| Biochemical incomplete response (BIR) | 23 | 12 | 11 | |
| Structural incomplete response (SIR) | | | | |
| Indeterminate response (IR) | | | | |
| Lost to follow-up | 4 | 3 | 1 | ND |

ND: Not determined.

[a]: One year follow-up.

[b]: Comparison between PTC patients with any of the studied somatic alterations *vs* patients without alterations.

[c]: *p* value combining Classic/Follicular *vs* Non-classic/Follicular.

[d]: FDR adjusted *P* value after multiple comparison correction.

The most commonly fused gene was *RET*, in 6/17 rearranged PTC cases (35.3%), and the majority of these, 4/6 (66.7%), presented high initial risk. Moreover, 1 high risk case also underwent a previous ALL without radiation treatment 9 months before PTC. All *RET* patients presented lymph node metastasis and 83.3% further invaded the lymph node vasculature. Additionally, the single poorly differentiated thyroid carcinoma case also harbored a *RET* translocation; making *RET* rearrangements present in a total of 7/58 tumors of the thyroid gland.

Rearrangements in *ALK* were the second most common genetic fusion followed by *NTRK* genes.

In the case of *NTRK* fusions, 4 cases were positive by pan-TRK IHC (Fig 2A and 2B) and subsequently positive for *NTRK3* rearrangements by FISH (Fig 2C). In independent FISH assays for *ETV6*, 1 of these cases was unequivocally positive for *ETV6* fusion (Fig 2D), assuming *NTRK3-ETV6* as the most probable fusion, one case was negative for fusions in *ETV6* and the remaining 2 cases did not provide a reliable diagnosis since the minimum of 100 tumor nuclei or the 5%



Table 2. Risk stratification and molecular markers.

| Histological classification | | Risk stratification | | | | | | | | Sanger Seq. | IHC + | | FISH + | | | | |
|---|---|---|---|---|---|---|---|---|---|---|---|---|---|---|---|---|---|
| Histologic type (n) | Variant (n) | ATA initial risk (n) | | | Dynamic risk stratification (DRS) [a] (n) | | | | | BRAF V600E (n) | BRAF V600E (n) | p-TRK (n) | NTRK3 (n) | ALK (n) | BRAF (n) | MET (n) | RET (n) |
| | | Low | Interme-diate | High | ER | IR | BIR | SIR | | | | | | | | | |
| Differentiated papillary thyroid carcinoma (57) | Classic (25) | 2 | 10 | 13 | 11 | 4 | 2 | 7 | 7 | 6 | 1 | 1 | 3 | 0 | 0 | 1 |
| | Follicular (14) | 1 | 4 | 9 | 10 | 1 | 0 | 3 | 0 | 0 | 2 | 2 | 1 | 1 | 1 | 2 |
| | Diffuse Scle-rosing (6) | 0 | 0 | 6 | 2 | 1 | 1 | 2 | 0 | 0 | 0 | 0 | 1 | 0 | 0 | 1 |
| | Solid/Tra-becular (1) | 0 | 0 | 1 | 0 | 0 | 0 | 1 | 0 | 0 | 0 | 0 | 0 | 0 | 0 | 0 |
| | Warthin-like (1) | 0 | 1 | 0 | ND | ND | ND | ND | 0 | 0 | 0 | 0 | 0 | 0 | 0 | 0 |
| | Oncocytic (1) | 0 | 0 | 1 | 0 | 0 | 1 | 0 | 0 | 0 | 0 | 0 | 0 | 0 | 0 | 1 |
| | Microcarci-noma (9) | 4 | 5 | 0 | 7 | 0 | 0 | 0 | 0 | 0 | 1 | 1 | 0 | 0 | 0 | 1 |
| Poorly differentiated thyroid carcinoma (1) | | 0 | 0 | 1 | 1 | 0 | 0 | 0 | 0 | 0 | 0 | 0 | 0 | 0 | 0 | 1 |

[a]Only patients with more than one year of follow-up were considered for DRS evaluation; ER: Excellent response; BIR: Biochemical incomplete response; SIR: Structural incomplete response; IR: Indeterminate response; ND: not determined.

of positive tumor cells cut-off value could not be reached. These 2 latter cases were assessed with the Oncomine Focus Assay NGS panel and *ETV6* was finally confirmed as the partner gene translocated with *NTRK3* in both cases. It is worth mentioning that the 4 cases with *NTRK3* fusions were among those cases that exhibited additional neoplasia: two high initial risk cases with a synchronous tumor, a melanoma or a solid pseudopapillary pancreatic neoplasm; and two intermediate risk cases with a previous medulloblastoma, which also received neck radiotherapy, and an adrenal neuroblastoma.

In addition, the poorly differentiated thyroid carcinoma presented faint cytoplasmic pan-TRK staining in less than 5% of isolated tumor cells by IHC and negative FISH results for *NTRK3*, *NTRK1* and *NTRK2*. Given that this case was previously assumed as *RET* rearranged by FISH we sought to further assess, or rule out, the possibility of a double rearrangement. This diagnostic uncertainty was also solved by NGS, where targeted sequencing confirmed the absence of genetic alterations in *NTRK* genes, but confirmed *CCDC6* as the *RET* fusion partner.

By implementing this complexity-increasing molecular algorithm, we could reach a reliable molecular diagnosis in a total of 25/58 (43.1%) cases and, the remaining 33/58 (56.9%) cases were assumed as cases with undetectable genetic alterations.

## Genetic alterations, clinicopathological features risk and outcome

Clinical and pathological features at diagnosis were not significantly different between tumors that carried fusions and those that were fusion negative. When considered together, those PTC cases containing a gene fusion were not associated with a particular histologic variant (Classic/Follicular *vs* non- Classic/Follicular) (*P* = 1), tumor size (T1/T2 *vs* T3/T4) (*P* = 0.240), tumor focality (focal *vs* multifocal) (*P* = 1), lymph node invasion (*P* = 0.059) or distant metastases (*P* = 0.109). In a similar way, no difference in the clinical and pathological features were disclosed between those cases with BRAF

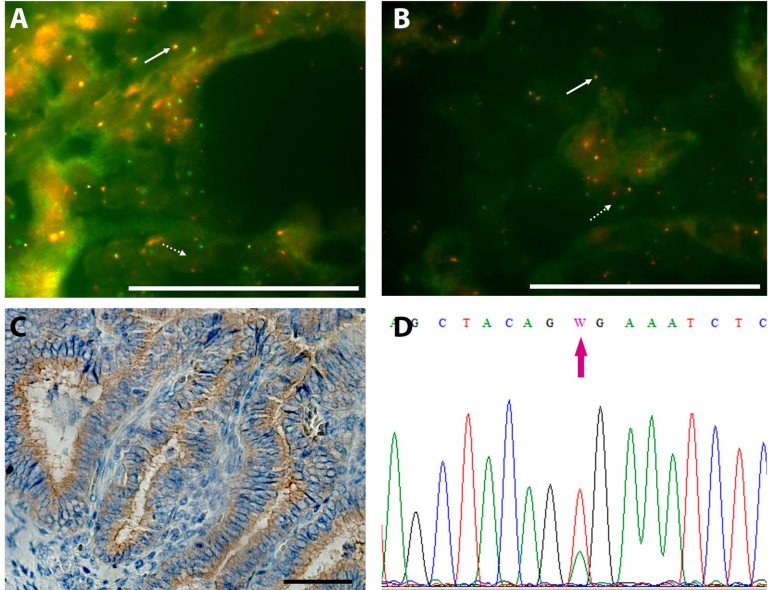

**Fig 1. Gene fusions and SNV detection in PTC.** (A) Representative case of an *RET* rearranged gene with break apart probes by FISH in a Follicular variant PTC. Full arrow shows a normal *RET* gene (yellow signal) and the dotted arrow shows a rearranged *RET* gene (green and red separated signals). **(B)** Representative case of an *ALK* rearranged gene with break apart probes by FISH in a Follicular variant PTC. Full arrow shows a normal *ALK* gene (yellow signal) and the dotted arrow shows a rearranged *ALK* gene (green and red separated signals). Scale bars represent 50 μm at 1000x magnification. **(C)** Representative immunohistochemistry for BRAF V600E mutation in a Classic variant of PTC. Scale bar represents 50 μm at 400x magnification. **(D)** Representative Sanger sequencing chromatogram showing double peaks in c.1799A>T (p.V600E), indicating a BRAF V600E heterozygous mutation. Magenta arrow shows the double peak.

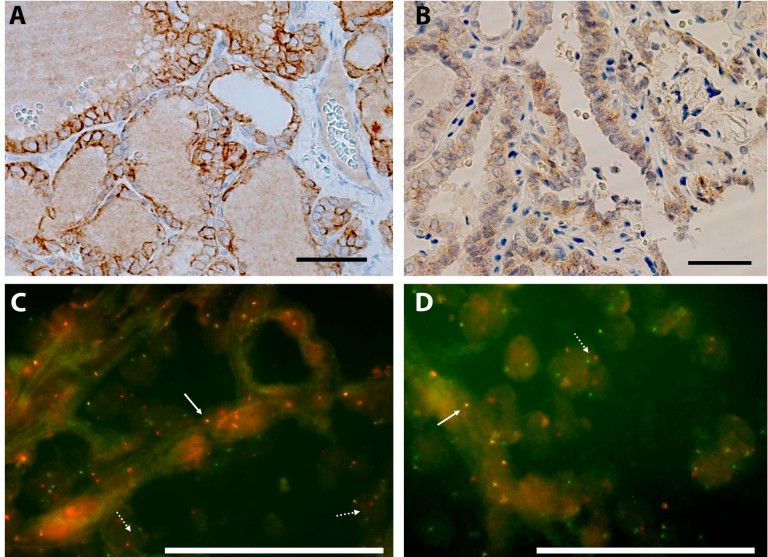

**Fig 2. Detection of *NTRK* alterations. (A and B)** Representative case of positive immunohistochemical staining for pan-TRK in a follicular variant of PTC. Scale bar represents 50 μm at 400x magnification. **(C)** Follicular variant PTC with *NTRK3* rearranged gene by FISH with break apart probes. Full arrow shows a normal *NTRK3* gene (yellow signal) and the dotted arrows show rearranged *NTRK3* genes (green and red separated signals) **(D)** Follicular variant PTC with *ETV6* rearranged gene by FISH with break apart probes. Full arrow shows a normal *ETV6* gene (yellow signal) and the dotted arrow shows a rearranged *ETV6* gene (green and red separated signals). Scale bars represent 50 μm at 1000x magnification.

V600E variant and cases without this SNV (P > 0.05 in all cases). When considering the combination of all genetic alterations, gene fusions and V600E SNV, these were still not associated with histologic variants (P = 0.161), tumor focality (P = 0.592), lymph node invasion (P = 0.160) or distant metastasis (P = 0.470); however, tumor size (T1/T2 vs T3/T4) was significantly larger in those PTC with genetic alterations (P = 0.027). Finally, gene fusions and/or V600E were not associated with thyroid autoimmune disease at onset of PTC (P = 0.420). When performing a multivariate logistic regression analysis, significance was retained for genetic alterations and tumor size; moreover, the OR of having a larger tumor was 3.49 times more likely when having a genetic alteration. This multivariate analysis reinforces the previous association between these variables.

Regarding initial risk assessment, 50 cases were categorized as high or intermediate risk. Of these, 4/50 harbored *NTRK3* fusions, 5/50 in *ALK*, 6/50 in *RET*, 1/50 in *MET*, 1/50 in *BRAF* and 7/50 had the BRAF V600E SNV. In total, 24/50 high/intermediate risk cases contained some genetic alteration. In consequence, the initial risk assessment (high/intermediate vs low risk) was statistically higher for cases that harbored any kind of genetic alteration (P = 0.036 with an OR = 3.00 in the multivariate analysis) (Table 1); but, when considering fusions or V600E SNV as separate categories, no significant association was revealed (P > 0.05 in both cases). Additionally, of the above-mentioned 50 cases, 48 could be further assessed for 1 year dynamic risk. Of these, 25/48 achieved ER status, while the remaining 23/48 cases were classified either as IR, SIR or BIR. Patients achieving ER status harbored genetic alterations in 12/25 cases: 2 with V600E SNV, 3 with fusions comprising *NTRK3*, 3 in *ALK*, 2 in *RET*, 1 in *BRAF* and 1 in *MET*. On the other hand, from the 23 non-ER cases, 11/23 contained a genetic alteration: 4 with V600E SNV, 1 with fusions comprising NTRK3, 2 in *ALK* and 4 in *RET*. In consequence, achieving an excellent response to treatment (ER vs non-ER) was not associated with the presence of gene fusions and/or V600E SNV (P > 0.05 in all cases) (Table 1).

## Discussion

This study aimed to investigate the frequency of genetic alterations and their associations with clinical and histopathological features, as well as with risk stratification, in a cohort of pediatric patients with PTC carcinoma from Argentina. Consistently with previous knowledge, 98.3% of our pediatric cases were differentiated carcinomas of the papillary histologic type [2,21,22] and only 1 case represented a poorly differentiated carcinoma. In this way, low risk variants (classic and follicular) represented 67.2% of our series, in line with that reported for other geographies and high risk variants (diffuse sclerosing, solid/trabecular and poorly differentiated carcinomas) accounted for 13.8% of the present series, almost alike the 15–37% described by Paulson et al. ([22] and references herein).

Since recent years, molecular assessment of PTC for gene fusions and SNVs is emerging as a complementary opportunity for risk stratification and second line treatment selection and clinical management [1,10]. However, and although several reports are available from Europe, Asia and North America [13,15,22,23], they describe small cohorts of varying ages, a fact that results in a wide range of described gene fusion and SNVs frequencies. In this way, in pediatric populations, fused genes have been described in a wide range of 14–71% of sporadic PTC cases ([22] and references herein), in line with our observed 29.8% of cases with gene fusions.

Among the characterized gene fusions in our PTC pediatric cohort, those involving *RET* (10.5%) were the most frequent, followed by fusions in *ALK* (8.8%) and *NTRK3* (7%) genes. Although fusions involving *RET* gene fell within the reported range of 0–57% in pediatric PTC cases from other regions [24], its frequency in our cohort fell in the lower quarter of this range. On the contrary, while *ALK* fusions were less frequently reported (6.5%) in children from other regions, *ALK* fusions were more prevalent than those involving *NTRK3* gene in Argentina [2,10,15,25] (S2 Table). Multiple factors may account for discrepancies in the frequency of fused genes in different geographic regions, including differences in cohort age distribution, gender composition, genetic background, and tumor histological variants [15,24]. Argentina's population is largely composed of immigrants from various regions of the world, some of whom have admixed with the original population, potentially contributing to differences in fusion frequencies.

It is worth mentioning that the majority of gene fusions (41%), and particularly those involving *NTRK3* gene (50%), occurred in the follicular variant of the present cohort of pediatric PTC; a fact also reported by Pekova et. al. [25]. Additionally, despite being rarely described in pediatric PTC [2,25], single cases of *MET* and *BRAF* fusions were also assessed in two independent cases of follicular PTC variants in the present series. Finally, 2 papillary microcarcinoma cases with ATA's intermediate initial risk and lymph node invasion harbored each a *RET* and a *NTRK3* gene fusion, suggesting the tumorigenic driver potential of gene fusions since the early stages of thyroid carcinogenesis.

In relation to BRAF V600E point mutation, although the observed frequency in 12.3% of our PTC cases is lower than the 30% described in average [2,10,13,23], is similar to the frequency described by Chakraborty et. al. [26] in pediatric PTC and falls within the lower third of the total 0–63% range previously reported for this SNV [15,22]. The low frequency of this SNV, as compared to the above-mentioned studies, could be due to the high proportion of children less than 15 years of age in our series (70%), since it's been widely reported that the BRAF V600E SNV increases along the age spectrum [2]. Indeed, reports comparing this SNV frequency in pediatric and adult cases, often report a higher frequency of BRAF V600E in adult cases, even in coexistence with other SNVs, such as variations in *TERT* promoter [23]. BRAF V600E gained special research interest since it was proposed to be a useful marker to predict aggressive behavior; however a recent metadata analysis reported the lack of association with tumor size, multifocality, lymph node metastasis, extrathyroidal extension or vascular invasion; nonetheless, BRAF V600E was associated to distant metastasis [27]. However, other reports by Li et. al., and Henke et. al., concluded that this SNV had no impact in the short [28] or long term [29] outcome, respectively. Still, BRAF V600E retains its early diagnosis importance since specific inhibitors exist as a second-line treatment option for relapsing tumors.

In line with previous studies, our findings indicate that gene fusions are more prevalent than SNVs in pediatric PTC cases [2,10,15,22] (S2 Table). This observation could be due to a higher cellular plasticity in pediatric developing thyroid tissue, which may favor large-scale chromosomal rearrangements (e.g., *RET*, *NTRK*, *ALK*) that could lead to gene fusions. These fusions often create constitutively active kinases that activate entire oncogenic pathways (e.g., MAPK, PI3K), which are strong drivers of oncogenesis and require less, or no additional point mutations. In contrast, adult PTC, arise in a more differentiated, less plastic and mutation-prone environment. In this context, adults may have been exposed to more mutagens over time, accumulating point mutations that eventually may lead to malignancy [15].

Regarding aggressive clinicopathological features and risk association, these were not associated to either gene fusions or BRAF V600E SNV independently, probably given the size of the cohort, which nonetheless was not small in comparison to other previous studies [22,23,30]; however cohort size has to be recognized as a study limitation. Remarkably, although not significantly associated with increased risk, all cases with *RET* fusions presented worse clinicopathological features. When combined, gene fusion or V600E SNV, the occurrence of genetic alterations was associated with an enlarged tumor size and higher initial risk, as defined by the ATA. Pediatric PTC demonstrates a favorable survival rate; however, up to 20% of patients with involvement of lateral lymph nodes may experience recurrence. The factors contributing to recurrence are diverse and can be influenced by disease extent, patient or tumor biology, and the thoroughness of the surgical removal procedure [31]. Despite the higher rate of disease recurrence when compared to adults, overall survival is higher in pediatric PTC; indeed, the 5-year relative survival rate for PTC is 99% [15]. Our cohort attained 100% of overall survival and 56.6% of the patients achieved ER one year after surgery. Moreover, all 7 cases initially assessed as ATA low risk, were free from any genetic alteration and achieved ER within the first year of follow-up. The occurrence of any given genetic alteration was not associated with dynamic risk status after the 12 months of follow-up.

Nowadays, reaching a definitive diagnosis which may impact treatment decisions is becoming inconceivable without molecular assessment, specifically in progressive RAI-refractory diseases, children with distant metastasis, or cases that present with morbidly invasive disease prior to surgery. Indeed, tumors driven by *RET* or *NTRK1/3* fusions were described to present a more invasive behavior, including extensive nodal and distant metastasis [2]. Additionally, it was also suggested that in RAI-refractory cases, selective fusion directed therapy may restore radioiodine avidity and lead to tumor

response [32]. Molecular alterations in *BRAF*, *NTRK1/3* and *RET* are among those drug-actionable markers for which a systemic molecular therapy has been approved by the United States Food & Drug Administration (FDA) as a second-line treatment in pediatric patients with PTC [2]. Although panel-targeted NGS appears as the most obvious screening technique, its cost and turnaround time still makes it not accessible for all public hospitals from developing countries. On the other hand, while both IHC and FISH target unique molecules, they should still be seriously taken into account in situations where access to NGS is restricted.

Here we propose a molecular algorithm based on IHC, FISH and Sanger sequencing applied in a complexity-increasing sequential order (Fig 3). Initial screening by IHC for BRAF V600E and pan-TRK are quick and affordable techniques for the routine pathology laboratory. Given the potential for false-negative results with the V600E antibody in some cases, negative cases should be assessed for this SNV by Sanger sequencing; however, we still consider BRAF V600E IHC an accessible and useful initial screening technique. Those cases positive for pan-TRK IHC, should be tested by FISH for *NTRK3*, *NTRK1*, *NTRK2* gene fusions, in this sequential order given their gene fusion frequency. At this point it is worth discussing that a first screening with pan-TRK antibody is a suitable approach since this antibody showed 100% specificity for thyroid tumors, minimizing the risk of false negative results. However, the staining pattern of this antibody may vary based on tumor type and the fusion partner, thus requiring pathologists to be familiar with this fact [18]. Subsequently, all BRAF and pan-TRK IHC negative cases should be tested for *RET*, *ALK*, *MET* and *BRAF* fusions by FISH, since these are the most prevalent gene fusions in children PTC [10,14]. While sporadic pediatric PTC cases with two concomitant genetic alterations have been documented [30,33], the majority of these genetic changes is mostly mutually exclusive and concentrated in a select group of genes, which results in a relatively low level of mutations compared to other types of cancers [34]. In accordance with most previous reports [13], all genetic alterations proved to be mutually exclusive in our cohort. Consequently, and considering those 33 cases without any detectable genetic alterations, we were able to diagnose 55/57 cases of PTC with this algorithm and reduce the requirements of NGS to just 2/57 PTC and 1 poorly differentiated thyroid carcinoma. Regarding the latter, it is worth mentioning that false positive pan-TRK weak staining in

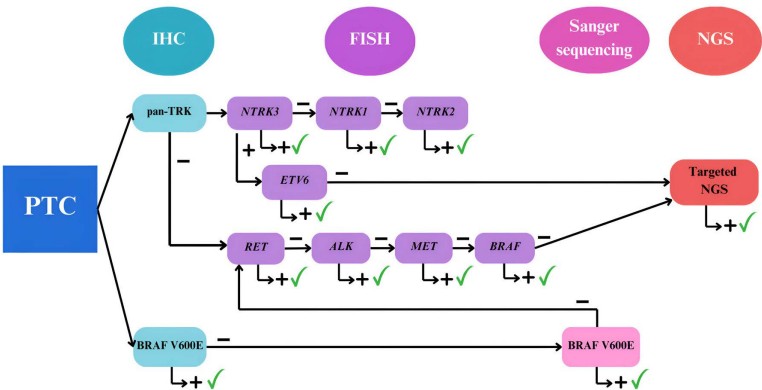

**Fig 3. Proposed molecular algorithm following a complexity-increasing order of techniques for the assessment of molecular alterations in PTC in a public hospital from a developing country.** All PTC samples should be initially screened for pan-TRK and BRAF V600E proteins by immunohistochemistry (IHC). Positive cases for pan-TRK should be sequentially tested for NTRK3, NTRK1 and NTRK2 gene fusions, following NTRK genes fusion frequency by fluorescent in situ hybridization (FISH). When a *NTRK3* fused gene is detected, a further FISH determination for *ETV6* should be performed, since *ETV6* is the most frequent fusion partner. BRAF V600E negative by IHC should be re-assessed by Sanger sequencing. All BRAF V600E and pan-TRK IHC negative cases should be sequentially tested for RET, ALK, MET and BRAF fusions, following the genes fusion frequency by FISH. Cases negative for all previous tests and those positive for *NTRK3* but negative for *ETV6* could be assessed by targeted next generation sequencing (NGS). Blue square refers to the PTC sample. Oval shapes represent different techniques and rounded-corner squares depict molecular targets. Teal color for IHC, violet for FISH, pink for direct Sanger sequencing and orange for NGS targeted sequencing. The green checkmark indicates an end-point in diagnosis.



*RET* rearranged tumors was previously described [35]. The proposed molecular algorithm proved to be very resourceful; however, and even though the scarce occurrence in pediatric cases of SNVs in other genes and the infrequent occurrence of a double genetic alteration [1], not assessing these other genetic alterations, or the possibility of concomitant alterations, could be considered as an algorithm´s limitation. In a similar way, sample-size and the single-center study design also represent study limitations; however the retrospective assessment of 57 pediatric PTC still represents a considerable sample given the low frequency of this tumor in children and the scarcity of data from South American series [10,15,22]. Future research studies with a larger and more diverse (multi-center) cohort of pediatric cases may prove the adaptability of the diagnostic algorithm to various healthcare environments in a widely diverse country, such as Argentina.

The molecular hallmarks underlying pediatric PTC still remain not entirely decoded. Furthermore, obstacles stemming from the costs and availability of sophisticated high-throughput techniques, particularly in less developed areas, add to this complexity. Future validation of this algorithm with these high-throughput techniques will be necessary, however in the context of resource constrained countries, techniques such as IHC and FISH provide the best example of rapid bench-to-bedside research tools since they allow for the assessment of the most prevalent genetic markers in pediatric PTC in a more cost-effective, widely available, and easier to perform manner in a pathology laboratory in a public hospital. Gaining a more comprehensive understanding of how molecular markers can be effectively used for diagnosis, risk assessment, and treatment strategies, it is imperative to gather information from different populations from all geographies around the globe. In this context we provide further epidemiological knowledge on molecular markers from pediatric cases of PTC from Argentina. Evaluating molecular markers for cancer diagnosis and clinical management is crucial in today's context, particularly given the shifting paradigm from focusing solely on "tumor type" to emphasizing the significance of "molecular alterations present".

## Supporting information

**S1 Table.  Estimated Age-Standardized Incidence Rates (ASR) for thyroid cancer.** Both sexes, per 100.000 for representative countries around the globe.
(DOCX)

**S2 Table.  Molecular alteration in adult and pediatric PTC.**
(DOCX)

**S3 Table.  FISH probe description.**
(DOCX)

## Acknowledgments

We thank Ms. Veronica Lapido for her technical assistance in the laboratory.

## Author contributions

**Conceptualization:** Mario Alejandro Lorenzetti, María Victoria Preciado.

**Data curation:** Marisa Esther Boycho, Patricia Papendieck, Mercedes García Lombardi, Elena Noemí De Matteo, Mario Alejandro Lorenzetti, María Victoria Preciado.

**Formal analysis:** Sandra Lorena Colli, Marisa Esther Boycho, Patricia Papendieck, Ana Chiesa, Mercedes García Lombardi, Martín Medín, Elena Noemí De Matteo, Mario Alejandro Lorenzetti.

**Funding acquisition:** Mario Alejandro Lorenzetti, María Victoria Preciado.

**Investigation:** Sandra Lorena Colli, Marisa Esther Boycho, Patricia Papendieck, Mario Alejandro Lorenzetti, María Victoria Preciado.



**Methodology:** Sandra Lorena Colli, Marisa Esther Boycho, Patricia Papendieck, Franco Mauricio Mangone, Ana Chiesa, Mercedes García Lombardi, Martín Medín.

**Project administration:** María Victoria Preciado.

**Resources:** María Victoria Preciado.

**Supervision:** Elena Noemí De Matteo, Mario Alejandro Lorenzetti, María Victoria Preciado.

**Validation:** Sandra Lorena Colli, Marisa Esther Boycho.

**Visualization:** Sandra Lorena Colli, Marisa Esther Boycho, Franco Mauricio Mangone, Ana Chiesa, Mario Alejandro Lorenzetti.

**Writing – original draft:** Sandra Lorena Colli, Marisa Esther Boycho, Mario Alejandro Lorenzetti, María Victoria Preciado.

**Writing – review & editing:** Patricia Papendieck, Franco Mauricio Mangone, Ana Chiesa, Mercedes García Lombardi, Martín Medín, Elena Noemí De Matteo, Mario Alejandro Lorenzetti, María Victoria Preciado.

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
