## [Decision Letter · Decision Letter 0]

8 Jan 2025

PONE-D-24-37301Prevalent genetic alterations in pediatric thyroid carcinoma: insights from an Argentinean studyPLOS ONE

Dear Dr. Preciado,

Thank you for submitting your manuscript to PLOS ONE. After careful consideration, we feel that it has merit but does not fully meet PLOS ONE’s publication criteria as it currently stands. Therefore, we invite you to submit a revised version of the manuscript that addresses the points raised during the review process. Please submit your revised manuscript by Feb 22 2025 11:59PM. If you will need more time than this to complete your revisions, please reply to this message or contact the journal office at plosone@plos.org . Please include the following items when submitting your revised manuscript:

We look forward to receiving your revised manuscript.

Kind regards,

Avaniyapuram Kannan Murugan, M.Phil., Ph.D.

Academic Editor

PLOS ONE

**Journal Requirements:**

This study was supported in part by a grant from the National Institute of Cancer (INC) of the National Ministry of Health (Res 83/2020) and grants from the National Agency for Research Promotion, Technological Development and Innovation (Agencia I+D+I) of the National Ministry of Science, Technology and Innovation (PICT 2020 Nº 854) and (PICT 2021 Nº 121). E.N.DM, A.C., M.A.L, and M.V.P are members of the CONICET Research Career Program. M.E.B was supported by a PhD fellowship from Agencia I+D+I.

**Additional Editor Comments:**

Despite the study is interesting, I and other invited reviewers feel it needs substantial revision prior considering it for publication. Moreover, the study overlooked many important findings in the field of pediatric thyroid cancer. Kindly look into the following articles in the field to include, PMID: 31289610; PMID: 30272236; PMID: 27824297; PMID: 26711586. In addition, consider carefully addressing all the reviewers' comments seen below.

Reviewers' comments:

Reviewer's Responses to Questions

**Comments to the Author**

1. Is the manuscript technically sound, and do the data support the conclusions?

Reviewer #1: Partly

Reviewer #2: Yes

2. Has the statistical analysis been performed appropriately and rigorously? 

Reviewer #1: No

Reviewer #2: I Don't Know

3. Have the authors made all data underlying the findings in their manuscript fully available?

Reviewer #1: No

Reviewer #2: Yes

4. Is the manuscript presented in an intelligible fashion and written in standard English?

Reviewer #1: Yes

Reviewer #2: Yes

5. Review Comments to the Author

**Reviewer #1:**  Is the manuscript technically sound, and do the data support the conclusions? Partly

Rationale:

- The methodology is generally sound with appropriate techniques

- However, several limitations affect the strength of conclusions:

* Small sample size without power calculation

* Single-center design limiting generalizability

* Lack of validation cohort

* Limited statistical rigor

* Overstatement of conclusions relative to data presented

Has the statistical analysis been performed appropriately and rigorously?No

Rationale:

- Basic statistical approaches only (chi-square, Fisher's exact, t-test)

- No correction for multiple comparisons

- Absence of multivariate analysis

- No power calculation

- Limited consideration of confounding variables

The statistical methodology demonstrates significant limitations that require addressing. The authors rely solely on basic statistical approaches (chi-square, Fisher's exact, t-test) without accounting for multiple comparisons or conducting multivariate analyses. The absence of power calculations and limited consideration of confounding variables undermines the robustness of their findings. Statistical rigor could be improved by incorporating survival analyses for follow-up data and implementing appropriate corrections for multiple testing.

3. Have the authors made all data underlying the findings fully available?Partly

FISH Controls and Quality Control Issues

Missing Essential Controls:

No demonstration of negative control samples

No presentation of internal controls

No validation of probe specificity

No quality control metrics for FISH signals

Technical Problems:

Lines 417-439: The FISH methodology section doesn't mention control samples

No images showing:

Negative controls

Internal controls

Background signal levels

Signal-to-noise ratios

Critical Impact:

Cannot validate specificity of FISH results

Unable to assess false positive/negative rates

Questions reliability of fusion detection

Compromises reproducibility of findings

4. Is the manuscript presented in an intelligible fashion and written in standard English?Yes

Rationale:

- Clear organization

- Generally well-written

- Minor typographical errors

Typographical Errors Examples:

Line 82: "Within the pediatric group the incidence of TC"

Should be: "Within the pediatric group, the incidence of TC"

Missing comma for proper clause separation

Lines 88-89: "between 2000 and 2009[5]"

Should be: "between 2000 and 2009 [5]"

Missing space before citation

Line 75-76: text formatting inconsistencies in spacing after periods

- Some sections could be more concise

Verbose Introduction Section: Original (Lines 78-81): "Although the thyroid gland is not the leading cancer site worldwide, thyroid cancer (TC) is the most common malignancy of the endocrine system, and its incidence has been steadily increasing over the past decade in all age groups"

Could be: "Thyroid cancer (TC), the most common endocrine malignancy, shows increasing incidence across all age groups"

Redundant Statements: Lines 19-20: "† Both authors contributed equally to this work † Both authors contributed equally to this work"

Unnecessary repetition

Wordy Methods Description: Lines 406-411: Could be more concise by combining technical details

My final Thoughts

While the manuscript presents valuable data on pediatric thyroid cancer in Argentina, substantial revisions are required before publication. The technical foundation is sound, but statistical analyses and data presentation require significant enhancement. With appropriate revisions addressing these concerns, this work could make a meaningful contribution to the field of pediatric thyroid cancer research.

The manuscript presents a methodologically structured investigation of genetic alterations in pediatric thyroid carcinoma from an Argentinian cohort. While the technical approaches (IHC, FISH, Sanger sequencing, NGS) are appropriate and well-executed, the study's conclusions are partially compromised by several limitations. The sample size (57 cases) lacks power calculation justification, and the single-center design limits generalizability. Although the experimental procedures are rigorous with appropriate controls, the absence of a validation cohort and limited follow-up period weakens the strength of the proposed diagnostic algorithm.

Recommendation

The manuscript requires major revisions before publication. While it presents valuable data on pediatric thyroid cancer in Argentina, the statistical analysis and conclusions need substantial strengthening to meet PLOS ONE standards.

**Reviewer #2:**  This manuscript offers valuable insights into genetic alterations in pediatric thyroid carcinoma in Argentina and proposes a practical algorithm for diagnosis. While the study is commendable, there are a few recommendations to improve clarity, add depth to the analysis, and enhance visual presentation. Below are my suggestions to help refine the manuscript.

1. Comparison Between Pediatric and Adult PTC

The manuscript touches on differences in molecular profiles between pediatric and adult PTC but doesn’t fully explore the clinical implications. It would be helpful to expand on why gene fusions like RET and ALK are more common in pediatric cases. These differences could stem from unique aspects of developmental biology or tumor pathogenesis. Additionally, elaborating on how these findings influence treatment approaches, such as the prioritization of fusion-targeted therapies, would add meaningful context.

2. Regional and Environmental Influences

The study finds a lower frequency of RET fusions in this Argentinean cohort compared to global reports, but it doesn’t delve into potential reasons. Including a discussion on environmental factors, such as radiation exposure (if there are some previous studies), or regional genetic predispositions, could strengthen the analysis. By comparing these findings with international data (e.g., Table S2), the authors can emphasize the novelty of this study.

3. Molecular Algorithm

The molecular algorithm proposed in this manuscript is practical and suitable for resource-limited settings, but it may have limitations. For instance, rare or complex genetic alterations might be missed due to its reliance on sequential IHC and FISH testing. Acknowledging this limitation and suggesting future validation in larger or more diverse cohorts would enhance the discussion. Highlighting its adaptability to various healthcare environments would also strengthen its applicability.

4. Algorithm Figure Quality

The figure illustrating the molecular algorithm (Figure 3) could be improved for better clarity and presentation. Adding clear labels for each step, consistent color coding for different molecular testing methods, and arrows to guide the flow would make it more visually intuitive. Using a high-resolution image with detailed legends and annotations would further enhance its accessibility.

Recommendation: Kindly revise the algorithm figure by including a detailed legend that explains each step and using distinct icons or symbols to differentiate methods like IHC, FISH, and NGS.

5. BRAF V600E Frequency

The manuscript notes a lower frequency of BRAF V600E mutations in this cohort but doesn’t fully explain why. This could be linked to the younger age of the cohort, as this mutation is more prevalent in older populations. Expanding on this connection and discussing how it impacts early diagnosis and treatment strategies in pediatric PTC would add depth to the findings.

6. Language Improvements

Some sentences in the manuscript are overly complex and could be simplified for better readability. Below are a few recommended revisions:

Page 13, Line 254:

Current: "In this study we sought to address the occurrence and frequency of genetic alterations..."

Recommended: "This study aimed to investigate the frequency of genetic alterations and their associations with clinical and histopathological features."

Page 14, Line 288-289:

Current: "Overall, and also in line with previous reports, in our pediatric series gene fusions were more prevalent than SNVs"

Recommended: "In line with previous studies, our findings indicate that gene fusions are more prevalent than SNVs in pediatric PTC cases."

Page 15, Line 318:

Current: "Since V600E antibody may give false negative results in a minor proportion of cases..."

Recommended: "Given the potential for false-negative results with the V600E antibody in some cases."

Page 15, Line 329:

Current: "Although pediatric sporadic cases of PTC with two concomitant alterations have been reported...etc"

Recommended: "While sporadic pediatric PTC cases with two concomitant genetic alterations have been documented."

6. PLOS authors have the option to publish the peer review history of their article (what does this mean? ). If published, this will include your full peer review and any attached files.

**Do you want your identity to be public for this peer review?** For information about this choice, including consent withdrawal, please see our Privacy Policy .

Reviewer #1: No

Reviewer #2: No

---

## [Author Response · Author response to Decision Letter 0]

6 Mar 2025

Rebuttal Letter

Below you will find a detailed point by point response to each comment raised by Reviewers, Academic Editor and PLoS Journal Editorial team.

Response to revision comments

Response: All issues raised by the Journal Editorial team were addressed:

- The manuscript was checked to meet PLOS ONE style requirements and file naming was corrected.

- The funding statement was corrected and the phrase “There was no additional external funding received for this study” was added. You can find the new statement here below and it was also included in the Cover Letter.

This study was supported by a grant from the National Institute of Cancer (INC) of the National Ministry of Health (Res 83/2020), grants from the National Agency for Research Promotion, Technological Development and Innovation (Agencia I+D+I) of the National Ministry of Science, Technology and Innovation (PICT 2020 Nº 854) and (PICT 2021 Nº 121) and a grant from Fundación Hector A. Barceló, year 2021-2022. There was no additional external funding received for this study. E.N.DM, A.C., M.A.L, and M.V.P are members of the CONICET Research Career Program. M.E.B was supported by a PhD fellowship from Agencia I+D+I.

- Regarding data availability, we agree to make all data publically available before acceptance. In order to achieve this we changed the Data Repository where all background and raw data are available. Data is now available at our National Research Council Data Bank: Repositorio Institucional CONICET Digital following the link: http://hdl.handle.net/11336/254857).

- Additional Editor Comments:

“Kindely look into the following articles in the field to include, PMID: 31289610; PMID: 30272236; PMID: 27824297; PMID: 2626711586”.

Response: The suggested reports were included either in the “Introduction” section or discussed in the “Discussion” section and added to the bibliography.

- Reviewer #1 comments:

1- Is the manuscript technically sound, and do the data support the conclusions? Partly

Rationale:

- The methodology is generally sound with appropriate techniques

- However, several limitations affect the strength of conclusions:

* Small sample size without power calculation

* Single-center design limiting generalizability

* Lack of validation cohort

* Limited statistical rigor

* Overstatement of conclusions relative to data presented

Response: In our report, the achieved statistical power is 70%. Standard studies aim for 80% statistical power to reduce the risk of Type II errors; however, in certain areas, such as pediatric oncology, a 70% power can be acceptable when sample size constraints exist. Given the rarity of pediatric thyroid carcinoma and the limited number of cases available, our study was constrained to a sample size of 57 cases. Consequently, the statistical power achieved was 70%, which, while lower than the widely used 80%, still provides meaningful insights into the association between genetic alterations and initial risk stratification. This study represents an exploratory analysis aimed at generating hypotheses for future research. While the power is slightly below the used threshold, our findings contribute valuable data to the limited literature on pediatric thyroid cancer in South America. Despite this, our analysis detected a statistically significant association between genetic alterations and larger tumor size and initial ATA risk (p =0.027 and 0.036 after correction, respectively), suggesting a meaningful clinical relationship. Although research reports don´t usually inform about statistical power, as indicated by Reviewer 1we included the following phrase at the end of the “Statistical analysis” section in “Materials and methods” (page 8, lines 279-280): “Given the sample size constraints, the achieved statistical power was 70%”.

We agree with Reviewer 1 that sample-size and the single-center study design limit generalizability and indeed, this was acknowledged in the Discussion section (page 21, lines 751-754). However, the retrospective assessment of 57 pediatric PTC still represents a considerable sample, especially when considering the low frequency of this tumor in children and given the scarcity of data from South American series. It should also be taken into consideration that we are one of the two reference pediatric hospitals in Argentina´s Capital City, where children arrive from different provinces for initial diagnosis, surgery and initial treatment, so we can argue that this cohort is representative from our nation and fairly represents the incidence of PTC in Argentina. Just as an example, consider that the review article by Cherella et. al. (PMID: 36404191) from 2022 reviews 7 research papers with a sample size that range from 27 to 93 cases, where 5 of these have less than 57 cases. Similarly, papers proposed to be included in the discussion by the Editor of this revision, include 55 pediatric cases (PMID: 26711586), a further enlarged cohort to 89 pediatric cases (PMID: 27824297), 20 pediatric cases with lung metastasis (PMID: 30272236) and a combination of the same 89 pediatric cases with adult cases (PMID: 31289610). In this context, our series is not so different and the study aims to describe the occurrence of the most frequent genomic alterations in pediatric thyroid cancer, which is why we paid special consideration for gene fusions. This study was not designed as a case-control study or an association study aimed to discover new molecular drivers of thyroid cancer, instead we aimed to describe the occurrence and the frequency of the most prevalent genetic alterations in children from Argentina with Thyroid Cancer and further we propose a molecular algorithm, based on the frequency of these molecular markers to facilitate the detection of these alterations in a rapid and more economical way in our limited resource setting.

Finally, since we are not proposing a new diagnostic method or test, but rather a sequence of already validated methods tailored to the frequency of genetic alterations in our population, we believe that a validation cohort is not necessary.

Other statistical concerns are addressed in the following point.

2- Has the statistical analysis been performed appropriately and rigorously?No

Rationale:

- Basic statistical approaches only (chi-square, Fisher's exact, t-test)

- No correction for multiple comparisons

- Absence of multivariate analysis

- No power calculation

- Limited consideration of confounding variables

The statistical methodology demonstrates significant limitations that require addressing. The authors rely solely on basic statistical approaches (chi-square, Fisher's exact, t-test) without accounting for multiple comparisons or conducting multivariate analyses. The absence of power calculations and limited consideration of confounding variables undermines the robustness of their findings. Statistical rigor could be improved by incorporating survival analyses for follow-up data and implementing appropriate corrections for multiple testing.

Response: Power calculation issue was addressed in the previous point and included in the Materials and Methods section.

Regarding the correction for multiple comparisons, in our work, each hypothesis tested was pre-defined based on prior biological knowledge, and each comparison addressed an independent clinical question, rather than testing multiple predictors against the same condition; in this scenario, correction for multiple comparisons wouldn´t be necessary. However, given that some conditions, like ATA initial risk classification, tumor size, lymph node metastasis and distant metastasis are biologically related, we applied a partial correction to the most conceptually linked comparisons.

These would be:

• Genetic alterations vs. Initial ATA risk (low vs. intermediate/high); p= 0.018

• Genetic alterations vs. Tumor size (T1/T2 vs. T3/T4); p= 0.027

• Genetic alterations vs. Lymph node metastasis; p=0.161

• Genetic alterations vs. Distant metastasis; p=0.471

Since the goal of correction is to control false positives (Type I error) while preserving statistical power and given that lymph node metastasis and distant metastasis were already not significantly associated with genetic alterations, including them in the correction set would dilute the correction effect unnecessarily and the comparison would still remain not-significant.

In this way, we applied the Benjamini-Hochberg (FDR) correction method to the two significant comparisons, which resulted in:

For p = 0.018: (0.018×2)/1=0.036

For p = 0.027: (0.027×2)/2=0.027

So, p values in Table 1 were corrected for p=0.036 and 0.027 (remains the same). The table footnote was also amended and all corrections are visible in track-changes.

Additionally, the phrase “Benjamini-Hochberg false discovery rate (FDR) test for multiple comparison corrections was applied to statistically significant biologically related comparisons. No correction was applied to other comparisons, as each addressed independent clinical questions” was included in the “Statistical analysis” section in “Materials and methods” (page 8, line 276-278).

A multivariate analysis was performed, as suggested by the reviewer, to identify independent predictors and for potential confounding variables. After multivariate analysis, the statistical significance between having a genetic alteration and a larger tumor and a higher initial ATA-risk remained significant. This analysis reinforced our previous results with classic statistics and rejects the possible occurrence of confounding variables.

The multivariate analysis methodology was added to the “Materials and methods” sections: (pages 8 and 9, lines 280-299) “Multivariate logistic regression analysis was performed to account for potential confounding variables between the occurrence of genetic alterations and clinical variables such as, tumor size, initial ATA risk, histological variant, sex, and age at diagnosis. P-values were calculated using the Wald test and odds ratios (OR) were derived to quantify effect sizes. Calculations were run with statsmodels 0.14.4 library in python v3.10.”

Additionally, the following phrases were added to the “Results” section: (page 16, lines 512-516) “When performing a multivariate logistic regression analysis, significance was retained for genetic alterations and tumor size; moreover, the OR of having a larger tumor was 3.49 times more likely when having a genetic alterations. This multivariate analysis reinforces the previous association between these variables”, and (page 16, lines 521-522) “(P=0.036 with an OR=3.00 in the multivariate analysis)”.

Regarding survival analysis, in our series 100% of the patients survived 1 year after surgery for which an overall survival analysis would not be informative. Moreover, it has been published that longer periods of follow-up also reported non-significant results. As an example, a long term follow-up study by Nies M, et. al. (PMID: 33382403) demonstrate an overall 5-year survival rate of 98.5% in their small BRAF mutated cohort group, not significantly different from that of wt-BRAF. Moreover, longer follow-up time from the same study ranged from 0.8–65 years, and thus authors report a 20-, 25-, and 30-year overall survival rate at 93.5%, 90.6%, and 86.8%, respectively. Even more, in a meta-analysis by Kotanidou EP et. al. (PMID: 36980495), authors concluded that, in pediatric protocols, recording survival is of poor scientific interest since it is largely expected to reach the maximum (100%), as demonstrated in the reports from Mollen et al. (PMID: 34915753), Hardee et al. (PMID: 28521635) and Henke et al. (PMID: 24677749), all of whom reported 100% survival in up to 42 years of follow-up.

Finally, Kotanidou EP et. al. also pointed out that the population subject to “loss of follow-up” during transition to adult health care professionals also complicates the derived survival outcomes, as systematic, long-term follow-up of pediatric patients with PTC is prone to missing data, especially when research in conducted out of a registry.

On the other hand, disease free survival is usually affected by variables such as age-at-diagnosis and presence of distant metastasis and usually requires a minimum of 5 years of follow-up. In consequence, in order to perform a disease-free survival analysis considering the presence of genetic alterations would require this follow-up period, something not possible in our series since some a large proportion of the series has not reached 5 years of follow-up.

3- Have the authors made all data underlying the findings fully available? Partly

FISH Controls and Quality Control Issues

Missing Essential Controls:

No demonstration of negative control samples

No presentation of internal controls

No validation of probe specificity

No quality control metrics for FISH signals

Response: Representative photographs of positive and negative control cases were provided for each probe in a review-purpose only file which is also available at the “Repositorio Institucional CONICET Digital (http://hdl.handle.net/11336/254857) Data Bank”. Additionally, during routine examination of each FISH slide, the pathologist examines the margins of the tissue in search of non-tumor cells and confirms the presence of probe signals in normal chromosomes, as an internal control. However, as FISH should be observed with an epi-fluorescence microscope at 1000x magnification, it is not possible to capture in a single field image, both tumor and non-tumor tissue. Hence, we cannot provide an image covering tumor and non-tumor sections as internal control.

We believe that an image containing photos for controls could be confusing for the general clinical reader; however we leave it to the Editor`s discretion if it should be included as a supplemental Figure or just available at the Data Bank.

FISH assays parameters, regarding tissue fixation, tissue thickness, deparaffinization process, digestion, probe concentration, incubation temperature and time, all wash steps and counterstaining were standardized to reduce the background fluorescence and increase the signal to noise ratio. Moreover, each FISH tissue section is carefully visualized by a trained pathologist at 1000x magnification using a dual color pass filter (filter set 25) to visualize both probes at the same time and estimate their separation. In this way the photographs presented in the manuscript are not merges from different color channels extracted from various images, but the actual picture that the pathologist is seeing and evaluating. Please consider that numeric background corrections and/or signal-to-noise corrections are mandatory for computer-assisted FISH quantification (or any other fluorescence quantifications), but not for qualitative FISH interpretation by a trained pathologist.

Finally, concerning our laboratory internal probe validation, we followed recommendation of the “Technical Standards for Clinical Genetics Laboratories (2024 Revision)” guidelines by the American College of Medical Genetics and Genomics (ACMG). Within these, regarding FISH probe validations, this guideline refers to an article by Mascarello JT et. al., PMID:21738013. In this article, the authors state that regular probes should be validated in order to confirm that they bind their specific target. They propose three methods to follow probe hybridization: i) DAPI counterstaining, ii) sequential G-/R-/ or iii) Q-banding. However, when referring to break-apart or fusion probes designs; a sample known to contain the abnormality of interest could also be used. The latter is exactly the validation process employed in our lab for our break-apart probes. Moreover, the latter approach has the advantage of also confirming the probe set´s ability to detect the abnormality and the advantage of confirming localization at the molecular level (gene locus), rather than the chromosomal region level.

In order to increase clarity and confidence level in our FISH technique for the general reader, the following phrase was added to the “Materials and Methods” section (page 7, lines 242-243): “All probes were internally validated using two tissue slides which were known to contain the specific abnormality”.

Finally, and regarding our l

---

## [Editor Report · Decision Letter 1]

6 Apr 2025

Prevalent genetic alterations in pediatric thyroid carcinoma: insights from an Argentinean study

PONE-D-24-37301R1

Dear Dr. Preciado,

We’re pleased to inform you that your manuscript has been judged scientifically suitable for publication and will be formally accepted for publication once it meets all outstanding technical requirements.

Kind regards,

Avaniyapuram Kannan Murugan, M.Phil., Ph.D.

Academic Editor

PLOS ONE
---

## [Editor Report · Acceptance letter]

PONE-D-24-37301R1

PLOS ONE

Dear Dr. Preciado,

I'm pleased to inform you that your manuscript has been deemed suitable for publication in PLOS ONE. Congratulations! Your manuscript is now being handed over to our production team.

Kind regards,

on behalf of

Dr. Avaniyapuram Kannan Murugan

Academic Editor

PLOS ONE